# THINK BEFORE YOU ACCEPT: SEMANTIC REFLECTIVE VERIFICATION FOR FASTER SPECULATIVE DECODING

## ABSTRACT

Large language models (LLMs) suffer from high inference latency due to the auto-regressive decoding process. Speculative decoding accelerates inference by generating multiple draft tokens using a lightweight model and verifying them in parallel. However, existing verification methods rely heavily on distributional consistency while overlooking semantic correctness, thereby limiting the potential speedup of speculative decoding. While some methods employ additional models for relaxed verification of draft tokens, they often fail to generalize effectively to more diverse or open-domain settings. In this work, we propose Reflective Verification, a training-free and semantics-aware approach that achieves a better trade-off between correctness and efficiency. Specifically, we leverage the inherent reflective capacity of LLMs to semantically assess the correctness of draft tokens in parallel during verification. Using prompt-based probing, we obtain both the original and reflective distributions of draft tokens in a single forward pass. The fusion of these distributions enables semantic-level verification of draft tokens that incorporates both consistency and correctness. Experiments across multiple domain benchmarks and model scales demonstrate that our method significantly increases the acceptance length of draft tokens without compromising model performance. Furthermore, we find that the proposed Reflective Verification is orthogonal to existing statistical verification methods, and their combination yields additional 5∼15% improvements in decoding speed.

## 1 INTRODUCTION

Large language models (LLMs), such as ChatGPT (Achiam et al., 2023) and Deepseek (Liu et al., 2024a), have demonstrated remarkable performance across a wide range of domains. However, they also face numerous challenges (Zhou et al., 2024b) during the deployment phase. One major contributor to the high inference latency of LLMs is the auto-regressive decoding mechanism inherent in decoder-only architectures. To mitigate the memory access bottlenecks associated with token-by-token generation, speculative decoding (Xia et al., 2024) has recently emerged as a promising approach for inference acceleration. This technique employs a lightweight draft model to propose multiple candidate tokens, which are then simultaneously verified by the target model.

Compared to the drafting stage, the primary objective of the verification stage is to determine whether the current candidate tokens are accepted. Using exact match as the verification criterion (Xia et al., 2023; Santilli et al., 2023) can indeed ensure the losslessness of the acceleration method. But it is often constrained in scenarios with high sampling temperatures. In order to further enhance the acceleration effectiveness of speculative decoding, several recent efforts (Kim et al., 2023) have focused on developing more relaxed verification strategies that increase the acceptance length of candidate tokens while preserving output correctness. Various statistical metrics have been employed to enhance the reliability of verification strategies. Speculative sampling (Leviathan et al., 2023; Chen et al., 2023) proposes an unbiased decoding strategy with respect to the original distribution of the target LLM, allowing flexible adjustment of the output based on the alignment between the draft and target distributions. In addition, several studies have explored the use of neural networks to decide whether candidate drafts should be accepted. Judge Decoding (Bachmann et al., 2025) trains a classifier on human-annotated data to achieve longer acceptance lengths without compromising

downstream task performance. Liao et al. (2025) introduces an auxiliary reward model to evaluate drafts, allowing for step-level discrimination of candidate tokens.

While the above methods improve the performance of speculative decoding during verification, two key challenges remain unresolved. **(1) The lack of semantic guidance.** Current mainstream approaches primarily perform verification using statistical information between the draft and target distributions, which provides limited information. There is a need for a more efficient verification mechanism guided by semantic-level information, where acceptance decisions are made based on semantic correctness rather than distributional consistency. **(2) Limited generalization.** Although some existing methods leverage deep models for verification, they typically require additional human annotations and training procedures. Moreover, many of these methods are tailored specifically for reasoning tasks with step-level responses, and thus struggle to generalize to more general scenarios.

In this paper, we propose **Reflective Verification**, a **training-free** draft verification method that operates at the **semantic level** to address the aforementioned challenges. Inspired by the observation in Figure 1 that self-reflection can effectively identify the semantic correctness of draft tokens, we leverage prompt-based probing to explicitly trigger reflection of the model within a single forward pass. The outcome of this reflection is then used to guide the verification of candidate draft tokens. Specifically, we exploit the unidirectional attention mechanism of LLMs by appending a reflection prompt and a copy of the draft tokens after original draft tokens during verification. This allows us to obtain the reflective judgment of target models on the current candidate tokens during the verification process. By integrating the original output representing consistency with the reflective output ensuring correctness, Reflective Verification can significantly extend the acceptance length of drafts while maintaining correctness, thereby improving the speedup of speculative decoding. In addition, the proposed method primarily calibrates the original output probabilities using reflective probabilities, which is orthogonal to existing statistical-based verification mechanisms. We conduct experiments across multiple configurations on benchmarks from various domains. The results demonstrate that the proposed method can bring orthogonal improvements to a wide range of existing verification strategies, achieving faster decoding without compromising task performance. Moreover, under low-quality draft settings, Reflective Verification helps mitigate the performance degradation of lossy verification methods and can even lead to overall performance improvements.

Our main contributions are as follows:

- We present a plug-and-play speculative decoding verification approach that incorporates semantic correctness by leveraging the reflective abilities of LLMs.
- By fusing the original and reflective outputs, the proposed method can be adapted to nearly all existing draft models and verification strategies, demonstrating strong generalization capability.
- Extensive results show that the proposed method can significantly extend draft acceptance length without degrading model performance, yielding a 5∼15% orthogonal improvement in end-to-end throughput.

## 2 OBSERVATIONS

In this section, we present several phenomena related to the verification of draft tokens that we have observed during the speculative decoding phase. Motivated by these observations, we further propose the Reflective Verification method.

### 2.1 NOT ALL REJECTED DRAFTS ARE INCORRECT

As discussed above, an effective verification mechanism requires a careful trade-off between correctness and inference efficiency. With the continuous advancement of small language models (Xiao et al., 2024), the quality of their draft tokens increases accordingly. Relying solely on strict consistency for draft verification can significantly limit the upper bound of speedup achievable by speculative decoding. In order to investigate potential improvements in verification strategies, we conduct an analysis of the draft tokens rejected by the standard speculative decoding verification process.

Figure 1 presents several examples of rejected draft tokens identified by the standard verification mechanism. We observe that some of these tokens, despite distributional inconsistencies, are se-

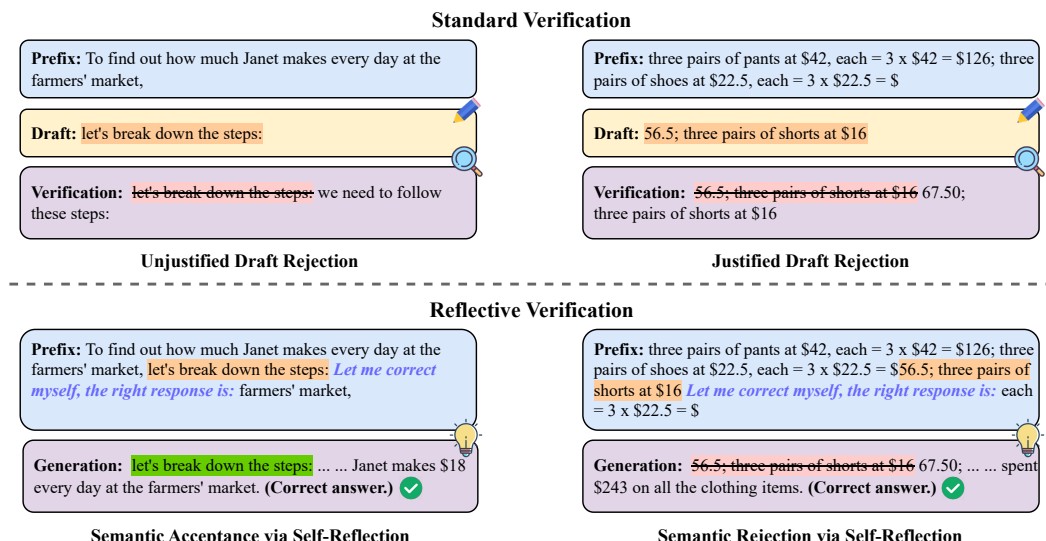

Figure 1: An illustration of draft tokens rejected by standard speculative decoding. Self-reflection enables the acceptance of semantically correct drafts that would otherwise be rejected.

mantically equivalent to the correct outputs. Accepting such tokens would not compromise the overall correctness of the response but can significantly improve the decoding speed. For example, in the case of unjustified draft rejection shown in the figure, although the token-level edit distance between the draft and the ground-truth output is large, the two sentences convey the same meaning. An effective verification strategy should accept such semantically correct drafts, thereby further improving the upper bound of speculative decoding.

Given this observation, we believe that current draft verification strategies remain suboptimal and will play an increasingly important role with the ongoing development of draft models. Developing a more relaxed and principled verification method is of great significance to the field of speculative decoding. In this paper, we explore how to leverage the reflection of LLMs to achieve semantic-level correctness rather than mere distributional consistency.

## 2.2 Self-Reflection Enables Correctness Verification

Although humans can naturally assess the correctness of draft tokens, verification mechanisms that depend only on statistical information from the draft and target model distributions often find it difficult to produce reliable decisions. Recent efforts (Bachmann et al., 2025) have aimed to achieve semantic-level speculative decoding by training classifiers using manually annotated draft acceptance labels. This approach typically requires additional annotation and training, and is difficult to quickly adapt to texts from other domains. Accordingly, we seek to explore training-free approaches that utilize the inherent capabilities of LLMs for semantic-level similarity verification.

Recently, self-reflection (Madaan et al., 2023; Ye et al., 2024; Chen et al., 2025) has garnered significant attention as a key property of LLMs. This behavior involves refining initially generated outputs by prompting the model itself through in-context learning (ICL). Motivated by this insight, we attempt to leverage the reflective behavior of LLMs to verify the semantic-level correctness of draft candidates. Specifically, we employ carefully designed prompts to induce the LLM to perform reflection and regeneration on the two rejected drafts discussed in Section 2.1.

As shown in Figure 1, we are surprised to find that, with reflective prompting, the LLM is capable of effectively distinguishing between draft candidates at the semantic level. In addition, unlike direct generation, the reflection process resembles an error-correction procedure applied to the original draft candidates, which aims to eliminate incorrect parts while preserving the original distribution as much as possible. This property of reflection makes it particularly suitable for use in speculative decoding verification, as it allows for the acceptance of a greater number of draft tokens while main-

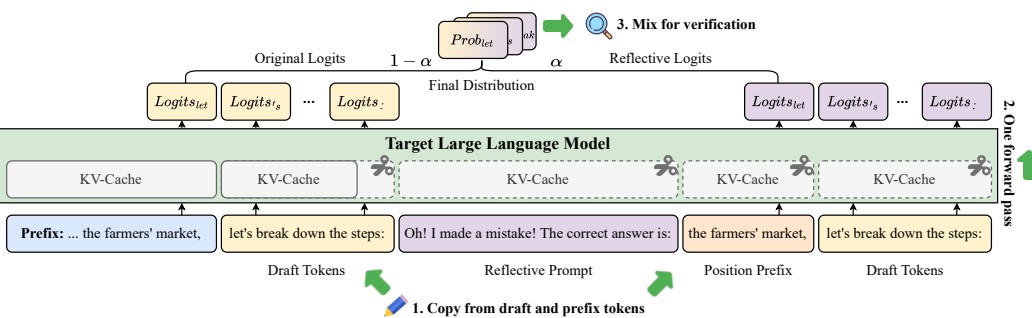

Figure 2: Overall structural diagram of Reflective Verification. Compared to vanilla speculative decoding using only base outputs (yellow), we fuse them with reflective outputs (purple) as the final distribution.

taining semantic correctness. Inspired by the above observations, we attempt to utilize the reflective output as auxiliary guidance to aid the original distribution in making more reliable verification decisions.

## 3  REFLECTIVE VERIFICATION

### 3.1  EXTRACTION OF REFLECTIVE LOGITS

The core of reflective verification lies in efficiently obtaining the reflection results of LLMs on draft candidates. While standard self-reflection is capable of evaluating the draft tokens, the reflection process itself still follows an auto-regressive decoding paradigm, making it impossible for direct application in speculative decoding. To address this, the proposed method employs prompt probing techniques to obtain two output distributions over the draft tokens in a single forward pass.

As shown in Figure 2, we apply a specialized design to the original draft tokens during the verification stage. Instead of directly feeding the draft tokens, we maintain two identical copies of the draft, with a reflective prompt probe inserted in between to explicitly trigger the reflection of LLMs. Benefiting from the unidirectional attention mechanism, the subsequent template leaves the verification of the initial draft tokens unaffected. The second draft tokens, informed by the context of probe, yields a reflection-based output that encodes semantic correctness verification. Specifically, the draft sequence constructed at each step of speculative decoding can be formulated as:

$$\text{Draft}_{\text{final}} = \text{Concat}(\text{Draft}_{\text{ori}}\|\text{Prompt}_{\text{reflection}}\|\text{Prefix}_{\text{position}}\|\text{Draft}_{\text{ori}}) \tag{1}$$

where $\text{Draft}_{\text{ori}}$ denotes the candidate tokens generated by the draft model, $\text{Prompt}_{\text{reflection}}$ is a probe designed to prompt the model to reflect, and $\text{Prefix}_{\text{position}}$ refers to the tokens preceding the current draft candidate within the context, serving to help the model locate the position for regeneration.

By feeding the constructed prompt into a single forward pass, we can efficiently obtain two distinct distributions over the draft candidate. Given the memory access bottlenecks during the decoding stage, the additional input does not significantly increase the forward latency. It is important to note that, except for the first draft segment, the KV-cache entries associated with other parts do not participate in subsequent computations. They are pruned after each forward pass, serving solely as a source of semantic-level verification signals.

### 3.2  FUSION OF ORIGINAL AND REFLECTIVE LOGITS

Despite sharing the same draft candidate tokens, the logits produced by the LLM for each draft segment carry different interpretations. The output of the first draft segment aligns with that of traditional speculative decoding and represents the distribution at the consistency level. By contrast, the second draft segment yields an output informed by the reflection of target LLM on the original draft, capturing the distribution corresponding to semantic correctness. To balance consistency with the original distribution and improved semantic correctness in the verification process, we fuse the

---

**Algorithm 1** Speculative Sampling with Reflective Verification

---

**Inputs:** $M_p, M_q, prefix, template, \alpha$.
**for** $i = 1$ **to** $\gamma$ **do**
    $q_i(x) \leftarrow M_q(prefix + [x_1, \ldots, x_{i-1}])$
    $x_i \sim q_i(x)$
**end for**
▷ Prepare the reflective draft template.
$reflective\_draft \leftarrow [x_1, \ldots, x_\gamma, template, x_1, \ldots, x_\gamma]$
$m \leftarrow \gamma + |template| + 1$
▷ Obtain the original logits $o_{1:\gamma+1}(x)$ and reflective logits $o_{m:m+\gamma}(x)$ in parallel.
$o_1(x), \ldots, o_{\gamma+1}(x), o_m(x), \ldots, o_{m+\gamma}(x) \leftarrow M_p(prefix), \ldots, M_p(prefix + reflective\_draft)$
▷ Fuse the two logits to obtain the final distribution $p_i(x)$.
$p_1(x), \ldots, p_{\gamma+1}(x) \leftarrow softmax((1-\alpha)o_1(x) + \alpha o_m(x), \ldots, (1-\alpha)o_{\gamma+1}(x) + \alpha o_{m+\gamma}(x))$
$r_1 \sim U(0, 1), \ldots, r_\gamma \sim U(0, 1)$
$n \leftarrow \min(\{i - 1 \mid 1 \le i \le \gamma, r_i > \frac{p_i(x)}{q_i(x)}\} \cup \{\gamma\})$
$p'(x) \leftarrow p_{n+1}(x)$
**if** $n < \gamma$ **then**
    $p'(x) \leftarrow norm(max(0, p_{n+1}(x) - q_{n+1}(x)))$
**end if**
$t \sim p'(x)$
**return** $prefix + [x_1, \ldots, x_n, t]$

---

original logits and the reflective logits to form the final output distribution of the target LLM. The final distribution for verification can be formulated as:

$$Prob_{mix}[i] = \text{Softmax}((1-\alpha) * Logits[i] + \alpha * Logits[i + \text{shift\_len}]) \tag{2}$$

We compute a weighted sum of the logits at position i and its corresponding reflective logits to obtain the final output distribution. shift_len denotes the number of tokens occupied by the designed prompt and the first draft segment, and $\alpha$ is a hyper-parameter that controls the weight of the reflective logits.

In essence, the proposed reflective verification mechanism uses the reflective logits as a side product to selectively align the distribution of target model with the semantically correct distribution produced by draft model. This method merely produces an output distribution with a higher acceptance rate and does not involve any specific verification mechanism. Therefore, it is fully orthogonal to existing statistical verification approaches and can be broadly applied across various draft models and verification settings.

### 3.3 SPECULATIVE DECODING WITH REFLECTIVE VERIFICATION

Once the reflective output distribution is obtained, the proposed method can be easily integrated with existing statistical verification approaches through minor modifications. To further illustrate how the proposed method can be applied, we take speculative sampling (Leviathan et al., 2023; Chen et al., 2023) verification as an example to present the overall algorithmic process. As demonstrated in Algorithm 1, the parts marked with green annotations denote the primary distinctions between reflective verification and standard speculative sampling. After obtaining the draft sequence, we construct the reflective draft for verification, which includes two copies of the draft tokens and a reflection prompt. By performing a single forward pass to compute the outputs of both copies in parallel and fusing them, we obtain the probability $p_i(x)$ of the target LLM, corresponding to that in standard method. Subsequently, all operations are identical to those in the standard speculative sampling verification mechanism.

It is worth noting that reflective process is fully decoupled from the draft generation and verification stages. This makes it compatible with nearly all draft generation and verification methods. See Appendix B for details on the integration of Reflective Verification with other statistical methods.

## 4 EXPERIMENTS

### 4.1 SETTING

**Benchmarks and metrics.**    In this paper, we select three commonly used benchmarks for different domains: MT-Bench (Zheng et al., 2023) for dialogue, GSM8K (Cobbe et al., 2021) for mathematics, and HumanEval (Chen et al., 2021) for code. We use the corresponding metrics as task performance indicators, with mean accepted tokens (#MAT) (Xia et al., 2024) for each forward pass and end-to-end throughput serving as speed metrics.

**Selected baselines.**    To demonstrate the generality of the proposed method, we conduct experiments across multiple draft model configurations and verification strategies. For the draft models, we choose two configurations from Llama3 series (Grattafiori et al., 2024) (1B&8B and 8B&70B) to investigate the impact of model scale on reflective verification.

As for verification strategies, we select the following three commonly used methods: (1) **Speculative Decoding** represents the naive lossless verification, in which verification is deemed successful only when the draft tokens exactly match the sampling of the target model. (2) **Speculative Sampling** (Leviathan et al., 2023) uses the probability ratio of candidate tokens under the target and draft distributions as the criterion, achieving unbiased verification through sampling. (3) **Typical Sampling** (Cai et al., 2024) relaxes the verification criterion by using an entropy-based threshold derived from the target distribution, significantly improving the acceptance rate of draft tokens.

**Generation config.**    Since all models used in the experiments are instruct versions, we perform generation in a zero-shot manner across all three datasets. Further details about hyperparameters, including $\alpha$, prefix length, and others, can be found in Appendix C. All experiments are conducted on a server equipped with two NVIDIA A100 GPUs (80GB each) and Intel(R) Xeon(R) Gold 6348 CPU @ 2.60GHz.

### 4.2 MAIN RESULTS

The main experiments are conducted across diverse settings, with the detailed results presented in Table 1. We apply the proposed method (Reflec Verify) to various statistical verification approaches under two draft model settings, and evaluate its impact on both task performance and acceleration performance. Overall, Reflective Verification is orthogonal to existing common verification methods. It can significantly increase the acceptance length of draft candidates, leading to improved end-to-end inference speed in speculative decoding. Notably, this improvement comes without significant task performance degradation, and may even enhance it in some cases.

**Acceleration performance.**    Initially, Reflective Verification consistently improves acceleration performance across various existing verification methods. Under a fixed draft length, it yields an acceptance length increase close to 1, with particularly notable gains in mathematics and code generation tasks. This leads to a 5~15% orthogonal improvement in end-to-end throughput, achieved in a training-free and plug-and-play manner. Moreover, the proposed method still brings improvements under typical sampling, which already achieves the highest acceptance length, demonstrating the broad applicability of our approach.

**Task performance.**    It is worth noting that Reflective Verification also brings certain improvements in task performance under lossy verification strategies. Although existing lossy verification strategies significantly increase acceptance length, they often introduce degradation in generation quality, which is particularly pronounced in objective tasks such as mathematics and code generation. The core issue is that distributional statistics alone cannot ensure the semantic correctness of draft tokens, often resulting in the acceptance of incorrect drafts. By leveraging the reflective signals of target LLMs, the proposed method enables semantic-level acceptance decisions for draft tokens. As shown in the table, incorporating semantic information not only increases the acceptance length but also effectively mitigates performance degradation. Moreover, incorporating Reflective Verification does not affect the overall output tokens length of target model. We further provide case studies in Appendix D to illustrate the semantic consistency brought by the proposed method.

Table 1: Main results across multiple benchmarks. Underline denotes performance degradation, and O.T. denotes output tokens length. **Bold** indicates the best result under each verification strategy.

| Method | MT-Bench | | GSM8K | | HumanEval | | Average | | | |
|---|---|---|---|---|---|---|---|---|---|---|
| | Score | #MAT | Acc. | #MAT | Pass@1 | #MAT | Perf. | O.T. | Tok./s | Speed |
| *Llama3.2-1B-Instruct & Llama3.1-8B-Instruct* | | | | | | | | | | |
| Vanilla AR | 7.44 | 1.00 | 77.63 | 1.00 | 65.85 | 1.00 | 72.63 | 477.52 | 45.96 | 1.00× |
| Spec Decoding | **7.44** | 3.43 | **77.63** | 6.12 | **65.85** | 6.68 | 72.63 | 480.88 | 51.02 | 1.11× |
| + Reflect Verify | 7.37 | **4.15** | 77.41 | **7.02** | **68.90** | **7.60** | 73.25 | 484.40 | **58.84** | 1.28× |
| Spec Sampling | **7.51** | 4.03 | **78.09** | 6.25 | 66.46 | 6.77 | 73.05 | 479.20 | 54.13 | 1.18× |
| + Reflect Verify | 7.44 | **4.88** | **78.09** | **7.15** | **69.51** | **7.65** | 73.92 | 472.67 | **62.32** | 1.36× |
| Typical Sampling | **7.65** | 4.82 | 76.57 | 6.76 | 63.41 | 7.27 | 71.88 | 476.96 | 60.19 | 1.31× |
| + Reflect Verify | 7.50 | **5.18** | **76.65** | **7.50** | **67.68** | **7.93** | 73.00 | 485.93 | **64.79** | 1.41× |
| *Llama3.1-8B-Instruct & Llama3.1-70B-Instruct* | | | | | | | | | | |
| Vanilla AR | 8.24 | 1.00 | 84.91 | 1.00 | 78.05 | 1.00 | 81.74 | 409.27 | 9.55 | 1.00× |
| Spec Decoding | 8.24 | 4.68 | 84.91 | 7.89 | 78.05 | 8.48 | 81.74 | 408.59 | 18.09 | 1.89× |
| + Reflect Verify | **8.34** | **5.93** | **85.06** | **9.33** | **78.66** | **9.52** | 82.33 | 411.63 | **20.00** | 2.09× |
| Spec Sampling | 8.32 | 5.82 | 85.57 | 8.07 | 78.66 | 8.65 | 82.43 | 415.39 | 19.86 | 2.08× |
| + Reflect Verify | **8.51** | **7.48** | **85.75** | **9.45** | **79.27** | **9.55** | 83.32 | 404.50 | **21.35** | 2.24× |
| Typical Sampling | 7.93 | 7.24 | **85.52** | 8.81 | 77.44 | 9.00 | 80.68 | 410.11 | 21.44 | 2.25× |
| + Reflect Verify | **8.17** | **7.94** | 84.91 | **9.80** | **80.49** | **9.86** | 82.35 | 417.55 | **22.72** | 2.38× |

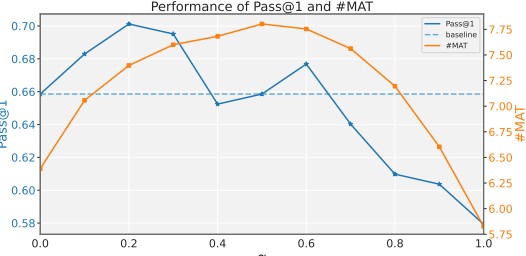

Figure 3: Effect of $\alpha$ on task and acceleration performance.

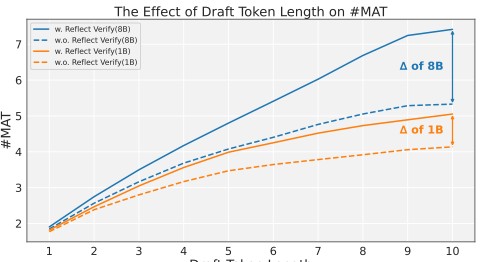

Figure 4: Impact of draft quality on Reflective Verification.

**Scale analysis.** To investigate the impact of reflective capability on the proposed method, we also conduct experiments under two different draft model configurations. While Reflective Verification provides improvements across different settings, its impact on performance is particularly significant in the 70B target model configuration. We attribute this to the fact that larger target LLMs possess stronger in-context learning and reflective capabilities, enabling them to make more informed judgments based on the provided prompts. This suggests that the proposed method holds great potential, with its effectiveness expected to improve as the scale and capabilities of LLMs increase.

## 5 ANALYSIS

### 5.1 THE TRADE-OFF IMPACT OF ALPHA

As a parameter controlling the weight of the reflective logits fusion, $\alpha$ plays a crucial role in the performance of the reflective verification process. To investigate how $\alpha$ balances consistency and semantic correctness in the reflective verification process, we conduct an ablation study on the hyperparameter. As shown in Figure 3, a trade-off relationship is observed between the value of $\alpha$ and the overall performance. When $\alpha$ increases from a low value, the growing influence of reflective logits leads to consistent and significant improvements in task performance and the number of accepted tokens. However, fully substituting the output distribution with reflective logits leads to performance degradation. Since the method is training-free, the reflective ability of LLMs is not reliable enough to maintain consistency with the original distribution. Based on the ablation results, we set the hyper-parameter to 0.3 in our experiments.

Table 2: Robustness of reflective prompts. Underline denotes performance degradation.

| Reflection Prompt | Pass@1 | #MAT |
|---|---|---|
| ${draft} | 65.85 | 6.39 |
| ${draft} Oh! I made a mistake! The correct answer is: ${prefix} ${draft} | **69.51** | **7.63** |
| ${draft} Let me correct myself, the right response is: ${prefix} ${draft} | 67.68 | 7.54 |
| ${draft} [BACK] ${prefix} ${draft} | 67.68 | 7.61 |
| ${draft} ${prefix} ${draft} | 64.63 | 7.45 |

## 5.2 IMPACT OF DRAFT QUALITY

As a verification approach, Reflective Verification does not directly enhance the quality of the draft itself. Instead, it aims to maximize the acceptance rate of semantically correct drafts under a given set of candidate drafts. Therefore, we conduct speculative decoding experiments using 1B and 8B draft models to investigate the effectiveness of the proposed method under varying draft quality conditions. Under the assumption that the 8B model generates higher-quality drafts, we evaluate the improvements brought by Reflective Verification over traditional methods across different draft lengths on the MT-Bench dataset.

As shown in Figure 4, the proposed method, by incorporating semantic information, significantly improves the acceptance rate at a fixed draft length, thereby raising the upper bound of speculative decoding performance. Notably, for the higher-quality drafts generated by the 8B model, Reflective Verification yields even greater improvements. This indicates that the verification mechanism does not merely increase the acceptance rate indiscriminately, but rather makes informed decisions based on the semantic correctness of the draft. In addition, with the advancement of draft models and improvements in draft quality, this semantics-level verification approach is expected to exhibit even greater potential.

## 5.3 ROBUSTNESS OF THE REFLECTIVE PROMPT

Reflective Verification leverages constructed reflective prompts as probes to elicit reflective capabilities from LLMs. The degree of sensitivity to these prompts directly influences the generalization capability of the proposed method. To evaluate the robustness of the proposed method to reflective prompts, we conduct experiments using a variety of alternative reflective prompts, as shown in Table 2. Specifically, the first row corresponds to standard speculative decoding without reflection, while the last row represents Reflective Verification with an empty reflective prompt.

It can be observed that the proposed method exhibits strong robustness to the reflective prompt. Whether using a full sentence or a simple token such as [BACK], it consistently outperforms standard speculative decoding. This demonstrates that the process of obtaining reflective logits by duplicating the draft tokens does not rely heavily on prompt engineering, allowing the method to be easily adapted to other settings.

## 5.4 COMPARISON WITH TREE-BASED VERIFICATION

As a method that also leverages additional input tokens to improve acceptance rates, tree-based verification (Miao et al., 2023) validates multiple candidate paths in a single forward pass by employing a sparse attention mask. Under a fixed input budget, we compare our method with the representative approach MCSD (Yang et al., 2024) on the MT-Bench dataset.

Table 3: A comparison with tree-based verification.

| Method | Config | #Budget | #MAT |
|---|---|---|---|
| Chain | {1x1x1x1x1} | 5 | 3.08 |
| MCSD | {4x2x2x1x1} | 60 | 4.06 |
| Ours | {5+3+4+5} | 17 | **4.92** |

The experimental results are shown in Table 3. For MCSD, the configuration denotes the number of nodes at each tree depth. For Reflective Verification, the configuration indicates the token counts of the four components in the prompt as defined in Equation 1. The reflective prompt used is "[BACK]". Experimental results show that when the draft already yields a fluent output (e.g.,

3.08 tokens accepted at chain mode.), the performance gains from traditional tree-based decoding become marginal. Moreover, as the depth increases, the tree structure consumes a larger portion of the available budget. In contrast, as draft generation models continue to improve, the benefits of reflective verification are expected to increase, offering greater potential.

# 6 RELATED WORKS

**Drafting methods of speculative decoding.** As a core step in speculative decoding, draft generation has attracted considerable attention from researchers (Xia et al., 2023; Bae et al., 2023; Liu et al., 2024b; Zhou et al., 2024a). To obtain more consistent information, some methods (Stern et al., 2018; Cai et al., 2024) have begun leveraging the hidden states of the target model for draft prediction. In particular, EAGLE (Li et al., 2024) demonstrates significant acceleration by training a standalone single-layer transformer designed to fuse token embeddings and the hidden states of the target model.

In contrast to approaches that rely on additional auxiliary models, some methods (Yang et al., 2023; Fu et al., 2024; Ou et al., 2024; Luo et al., 2024) aim to generate draft tokens more efficiently through retrieval-based techniques. REST (He et al., 2024) enables efficient draft tree construction and verification by building an index over the corpus. In addition, some studies explore parallel decoding (Santilli et al., 2023) to harness the capabilities of LLMs for self-drafting. CLLMs (Kou et al., 2024) improves the parallel decoding capability of LLMs by constructing and training on Jacobi decoding trajectories. Despite differences in draft generation, acceptance rates consistently improve with Reflective Verification by leveraging semantic signals.

**Verification methods of speculative decoding.** In addition to generating more consistent drafts, numerous studies (Chen et al., 2023; Leviathan et al., 2023) focus on improving verification methods to increase acceptance rates. Under lossless acceleration, increasing acceptance rates hinges on the ability to verify multiple drafts simultaneously. By utilizing sparse attention matrices, SpecInfer (Miao et al., 2023) accomplishes the verification of multiple draft paths in one forward computation, leading to a notable improvement in acceptance length. TR-Jacobi (Wang et al., 2024) achieves an orthogonal fusion of model-based and retrieval-based methods by incorporating retrieved paths into tree-based verification.

For lossy acceleration methods, the core lies in accepting as many inconsistent yet correct tokens as possible. Cai et al. (2024) select plausible candidates for acceptance using an entropy-dependent threshold. Qin et al. (2024) propose the multi-token joint decoding (MTJD), which performs verification based on the joint probability distribution rather than single token. Although some methods (Bachmann et al., 2025; Liao et al., 2025) leverage models with deep representations, they typically require additional models and training. We achieve a favorable balance between semantic-level validation and plug-and-play applicability.

# 7 DISCUSSION

**Limitations & Future.** This work does not explore Reflective Verification on a wider range of draft models or larger-scale models (e.g., 405B). While it significantly improves accepted draft length, it also increases step-wise variance, underscoring the need for dynamic draft length. For fairness and control, we adopt a fixed draft length in this study, and leave its dynamic adaptation to future work.

**Conclusion.** In this paper, we introduce Reflective Verification, a training-free, semantic-level verification method for speculative decoding. It is widely compatible with mainstream speculative decoding methods, boosting acceptance rates and enabling 5~15% faster decoding with no performance degradation. The proposed method shifts the verification criterion from exact consistency to semantic correctness, significantly raising the upper bound of speculative decoding and enabling the use of larger models as draft generators.

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

## A  USAGE OF AI TOOLS

In this paper, AI tools such as ChatGPT are used exclusively for grammar and proofreading of the final manuscript.

## B  DETAILS IN THE ALGORITHM

We present the pseudocode of the proposed Reflective Verification integrated with speculative de- coding and the typical sampling algorithm. As shown in Algorithms 2 and 3, only minimal mod- ifications are required to integrate Reflective Verification with existing statistics-based verification methods.

---

**Algorithm 2** Speculative Decoding with Reflective Verification

---

**Inputs:** $M_p, M_q, prefix, template, \alpha$.
**for** $i = 1$ **to** $\gamma$ **do**
  $q_i(x) \leftarrow M_q(prefix + [x_1, \ldots, x_{i-1}])$
  $x_i \sim q_i(x)$
**end for**
▷ Prepare the reflective draft template.
$reflective\_draft \leftarrow [x_1, \ldots, x_\gamma, template, x_1, \ldots, x_\gamma]$
$m \leftarrow \gamma + |template| + 1$
▷ Obtain the original logits $o_{1:\gamma+1}(x)$ and reflective logits $o_{m:m+\gamma}(x)$ in parallel.
$o_1(x), \ldots, o_{\gamma+1}(x), o_m(x), \ldots, o_{m+\gamma}(x) \leftarrow M_p(prefix), \ldots, M_p(prefix + reflective\_draft)$
▷ Fuse the two logits to obtain the final distribution $p_i(x)$.
$p_1(x), \ldots, p_{\gamma+1}(x) \leftarrow softmax((1-\alpha)o_1(x) + \alpha o_m(x), \ldots, (1-\alpha)o_{\gamma+1}(x) + \alpha o_{m+\gamma}(x))$
▷ Use exact match verification.
$\hat{x}_1 \sim p_1, \ldots, \hat{x}_\gamma \sim p_\gamma$
$n \leftarrow \min(\{i - 1 \mid 1 \le i \le \gamma, x_i = \hat{x}_i\} \cup \{\gamma\})$
$t \sim p_{n+1}(x)$
**return** $prefix + [x_1, \ldots, x_n, t]$

---

## C  HYPERPARAMETER DETAILS

For the main experiments in Table 1, we adopt the following hyperparameter settings. We present the configurations of speculative decoding and generation settings in Table 4. For different draft models and datasets, we select the optimal draft length K and temperature.

## D  DETAILED CASE STUDY

To further investigate the impact of Reflective Verification on model outputs, we conduct case studies on three datasets and present several representative examples. As shown in Figures 5 and 6, we segment semantically similar blocks between the two outputs. Despite variations in phrasing, the overall output length and semantic content remain consistent.

---

**Algorithm 3** Typical Sampling with Reflective Verification

---

**Inputs:** $M_p, M_q, prefix, template, \alpha, \epsilon, \delta$.
**for** $i = 1$ **to** $\gamma$ **do**
    $q_i(x) \leftarrow M_q(prefix + [x_1, \ldots, x_{i-1}])$
    $x_i \sim q_i(x)$
**end for**
$reflective\_draft \leftarrow [x_1, \ldots, x_\gamma, template, x_1, \ldots, x_\gamma]$
$m \leftarrow \gamma + |template| + 1$
▷ Obtain the original logits $o_{1:\gamma+1}(x)$ and reflective logits $o_{m:m+\gamma}(x)$ in parallel.
$o_1(x), \ldots, o_{\gamma+1}(x), o_m(x), \ldots, o_{m+\gamma}(x) \leftarrow M_p(prefix), \ldots, M_p(prefix + reflective\_draft)$
▷ Fuse the two logits to obtain the final distribution $p_i(x)$.
$p_1(x), \ldots, p_{\gamma+1}(x) \leftarrow softmax((1-\alpha)o_1(x) + \alpha o_m(x), \ldots, (1-\alpha)o_{\gamma+1}(x) + \alpha o_{m+\gamma}(x))$
▷ Use entropy-based threshold verification.
$threshold = \min\left(\epsilon, \delta \exp\left(-H\left(p_{\text{original}}(\cdot \mid x_1, x_2, \cdots, x_{n+k-1})\right)\right)\right)$
$n \leftarrow \min(\{i-1 \mid 1 \le i \le \gamma, p_i(x) > threshold\} \cup \{\gamma\})$
$t \sim p_{n+1}(x)$
**return** $prefix + [x_1, \ldots, x_n, t]$

---

Table 4: Details of the hyperparameters under different experimental settings.

| Setting | Dataset | Assistant K | $\alpha$ | Temperature | Prefix Len |
|---------|---------|-------------|----------|-------------|------------|
| 1B&8B | MT-Bench | 5 | 0.3 | 0.8 | 4 |
| 1B&8B | GSM8K | 8 | 0.3 | 0.2 | 4 |
| 1B&8B | HumanEval | 8 | 0.3 | 0.2 | 4 |
| 8B&70B | MT-Bench | 8 | 0.3 | 0.8 | 4 |
| 8B&70B | GSM8K | 10 | 0.3 | 0.2 | 4 |
| 8B&70B | HumanEval | 10 | 0.3 | 0.2 | 4 |

Notably, in the code generation domain, we observe that the performance gains from Reflective Verification primarily stem from improved handling of boundary cases. As shown in Figure 7, through self-reflection, the model becomes more sensitive to such edge conditions. By integrating reflective logits, it is able to generate higher-quality code.

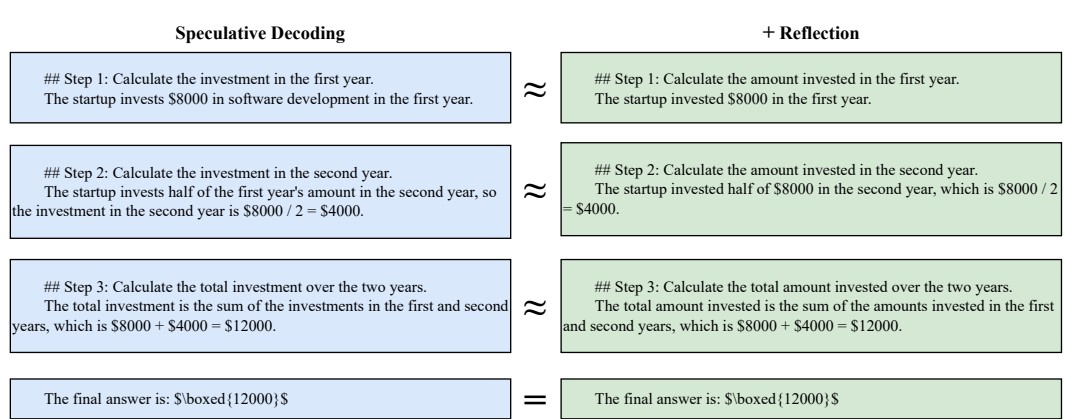

Figure 5: An illustration of reflective verification on MT-Bench.

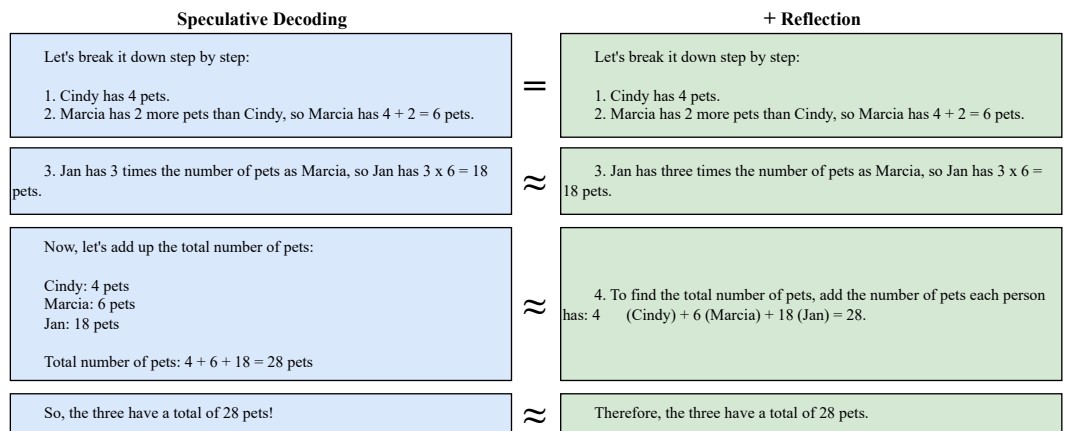

Figure 6: An illustration of reflective verification on GSM8K.

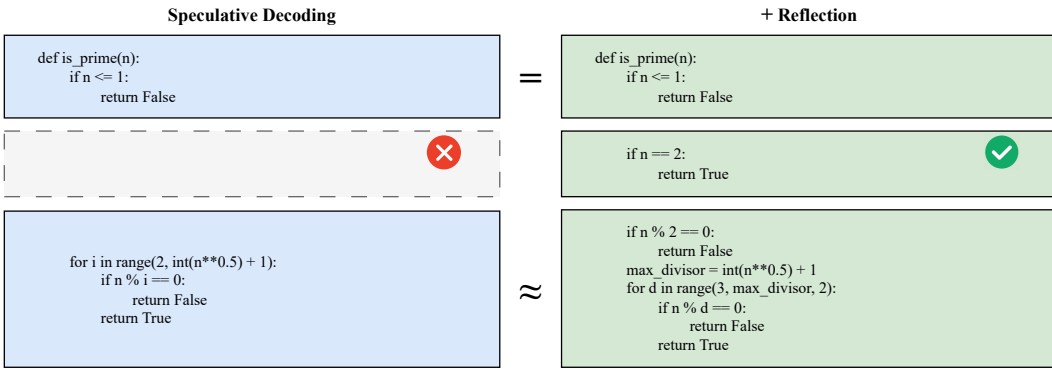

Figure 7: An illustration of reflective verification on HumanEval.

