# OpenReview forum: "Think Before You Accept: Semantic Reflective Verification for Faster Speculative Decoding"
_ICLR.cc/2026/Conference — ICLR 2026 Conference Withdrawn Submission_

### Official Review · Reviewer_XQTz · 2025-10-18

**Soundness:** 2
**Presentation:** 3
**Contribution:** 2
**Rating:** 4
**Confidence:** 4

**Summary:**

This paper introduces Reflective Verification, which relaxes the verification criterion in speculative decoding to account for semantically equivalent drafts that would have otherwise been rejected. Reflective Verification works by 1) repeating the draft twice in the target model's input for verification, with a "reflective prompt" in between that asks the LLM to correct itself; 2) mixing the logits of the two draft copies when determining whether to accept or reject the draft tokens.

**Strengths:**

The paper is well-written. The proposed method is conceptually simple and easy to understand.

**Weaknesses:**

- The original speculative decoding algorithms (https://arxiv.org/abs/2211.17192, https://arxiv.org/abs/2302.01318) are mathematically guaranteed to maintain the same output distribution as the target LLM. The proposed method, while empirically shown to be useful, is technically no longer a speculative algorithm and lacks a theoretical explanation.

- The method introduces multiple hyperparameters - the mixing weight $\alpha$, and the reflective prompt. These raise complexities in real-world deployment.

- The authors claim "Reflective Verification is compatible with nearly all draft generation and verification methods" (Line 269). However, state-of-the-art tree-based verification - such as EAGLE-2, one of the acknowledged fastest speculative decoding methods - is not discussed in the paper.

- The experimental setting is questionable.

  - In Appendix D, the authors mention "For different draft models and datasets, we select the optimal temperature". This is similar to "training on the test set" - temperature is a configuration parameter chosen by the user after deployment, not by the developer. So you cannot "choose" a temperature to report for each target-draft pair. I suggest reporting two sets of results, one in greedy decoding ($t=0$), and one in sampling (e.g. $t=0.6$).

  - In Section 4.1, the authors claim to use "three commonly used methods for verification". However, these three methods - especially the first two - are not parallel choices that you can arbitrarily apply. The so-called "speculative decoding with exact match" is only correct in greedy decoding, while the "speculative sampling" is meant for scenarios where temperature $t>0$.

**Questions:**

- The method introduces a notable amount of extra tokens in each verification step. While the authors claim "the additional input does not significantly increase the forward latency" in Line 204, I'm doubtful about that and would like to see empirical justification.

---

### Official Review · Reviewer_DmAq · 2025-10-29

**Soundness:** 2
**Presentation:** 3
**Contribution:** 3
**Rating:** 6
**Confidence:** 4

**Summary:**

Reflective Verification is a modification of the classic speculative decoding framework where semantic information is included in the target token distribution. The authors observe that under standard speculative decoding, semantically correct drafts can be rejected because the phrasing does not match the target model’s. Reflective verification thus fuses semantic information into the target model’s verification logits. This is achieved by appending a reflective prompt, a position prefix, and a copy of the draft tokens to the original input to the target model for verification. The output logits on the original draft tokens remain unchanged, but the logits on the copy of the draft tokens carry semantic correctness information. These logits are fused to produce a semantics-aware distribution to use for verification.

The semantic-aware distribution is different from the target model’s original distribution, meaning Reflective Verification is not a lossless speculative decoding method. However, the authors show that this loss incurs only negligible generation quality degradation. Across three domains spanning dialogue, math, and coding, and two draft/target model configurations, Reflective Verification demonstrates longer mean accepted tokens and faster generation speed at minimal task performance degradation.

**Strengths:**

- Obtaining the original and semantic logits in parallel in a single forward pass incurs little additional overhead compared to standard speculative decoding.
- The approach of fusing logits to modify the target distribution is (to my knowledge) novel, and seems to me to be an intriguing direction of further research.
- Strong empirical results demonstrate larger mean accepted tokens with minimal quality degradation.
- Important hyper-parameters are ablated and justified.

**Weaknesses:**

- Experiments are only conducted on the Llama 3 model family. Evaluations with other model families would strengthen the claims of the paper.
- I have minor concerns about the underlying mechanisms of the method. The prompt used to obtain the semantic logits includes the verified prefix, the newly drafted tokens, a reflective prompt, a position prefix (which is a suffix of the verified prefix), and the newly drafted tokens again. By the nature of LLMs, it seems the target model is heavily incentivized to output semantic logits that agree with the drafted tokens as the position prefix encourages the model to continue repeating by repeating the draft tokens. Successful verifications will increase, but it is less clear whether this is due primarily to the target model repeating the draft tokens or the draft tokens being actually semantically correct. More analysis on these effects (e.g., investigating how often semantically incorrect drafts are wrongly verified, does a worse draft model negatively affect performance) would strengthen the contribution of the paper.
The only baseline compared against is standard speculative decoding. There has been some work earlier this year that also aim to incorporate semantic information into draft verification (e.g., SpecReason (Pan et al., 2025), Speculative Thinking (Yang et al., 2025), Lookahead Reasoning (Fu et al., 2025)), though none to my knowledge directly modify the target token distribution. Still, these would be good baselines to compare against and clarify the benefits of the method
- A lack of confidence intervals on the results makes it difficult to contextualize the significance of the gains.

**Questions:**

- The reflective prompt seems to show up pretty suddenly in the input to the target model, potentially being appended when the draft model has not finished a sentence/word. Did you consider whether this could adversely affect the target model’s capabilities during reflection?
- For the draft quality ablation (Section 5.2), was the target model for both draft models the 70B model, or were these run with the original 1B/8B and 8B/70B configurations?
- For the tree-based verification ablation (Section 5.4), why was the reflective prompt used “[BACK]” and not the longer one used in the main experiments? Also, what is the “chain” baseline method?
- Minor writing nit: “Reflect Verify” rather than “Reflec Verify” in line 300.

---

### Official Review · Reviewer_rPhr · 2025-11-01

**Soundness:** 2
**Presentation:** 3
**Contribution:** 2
**Rating:** 4
**Confidence:** 4

**Summary:**

The paper proposes Reflective Verification, a training-free, semantics-aware method to enhance speculative decoding in Large Language Models. It leverages the model’s inherent reflective abilities by introducing prompt-based probing during verification, generating both original and reflective token distributions in a single forward pass. By fusing these outputs, the method improves semantic correctness and significantly increases draft token acceptance length, leading to faster inference without compromising performance. Compatible with most existing speculative decoding strategies, Reflective Verification delivers an additional 5–15% speedup and even helps mitigate performance drops in low-quality draft settings.

**Strengths:**

- Introduces a novel, training-free method that leverages LLMs’ self-reflective capabilities to assess semantic correctness, going beyond traditional distributional checks.
- Can be seamlessly integrated with existing speculative decoding and verification frameworks, demonstrating strong generalization across models and domains.
- Significantly increases draft token acceptance length and achieves an additional 5–15% decoding speedup without sacrificing model performance.

**Weaknesses:**

- The paper does not clearly define core terms like “semantic correctness,” making the approach harder to interpret and evaluate.
- The rationale behind fusing semantic and consistency-based token distributions lacks detailed explanation, leaving concerns about potential interference and effectiveness.
- The role and selection of key parameters (e.g., alpha) are not well explained, limiting the method’s reproducibility and adaptability across different models and tasks.

**Questions:**

- What do “semantic correctness” and “distribution of semantic correctness” mean exactly, since it is not rigorously defined in the paper?
- As for section 3.2, can you elaborate more on the motivation that fusing the distribution of the second draft segment (semantic) with the distribution of the first draft segment (consistency), can improve the acceptance rate qualitatively? (i.e., is it possible for the distribution of the second draft segment to interfere with  (disturb) the distribution of the first draft segment?)
- As for section 5.1, can you briefly explain why, in the training-free method, the reflective ability of LLMs is not reliable enough to maintain consistency with the original distribution?
- As for section 5.1, does the choice of alpha hyper-parameter change across different draft/target model selections? If so, how would you set alpha during inference time? If not, can you explain why the choice of alpha will not change in different setups?

---

### Note · Authors · 2025-11-25

I have read and agree with the venue's withdrawal policy on behalf of myself and my co-authors.